# Associative Memory via a Sparse Recovery Model

**Arya Mazumdar**
Department of ECE
University of Minnesota Twin Cities
arya@umn.edu

**Ankit Singh Rawat**[*]
Computer Science Department
Carnegie Mellon University
asrawat@andrew.cmu.edu

## Abstract

An *associative memory* is a structure learned from a dataset $\mathcal{M}$ of vectors (signals) in a way such that, given a noisy version of one of the vectors as input, the nearest valid vector from $\mathcal{M}$ (nearest neighbor) is provided as output, preferably via a fast iterative algorithm. Traditionally, binary (or $q$-ary) Hopfield neural networks are used to model the above structure. In this paper, for the first time, we propose a model of associative memory based on *sparse recovery* of signals. Our basic premise is simple. For a dataset, we learn a set of linear constraints that every vector in the dataset must satisfy. Provided these linear constraints possess some special properties, it is possible to cast the task of finding nearest neighbor as a sparse recovery problem. Assuming generic random models for the dataset, we show that it is possible to store super-polynomial or exponential number of $n$-length vectors in a neural network of size $O(n)$. Furthermore, given a noisy version of one of the stored vectors corrupted in near-linear number of coordinates, the vector can be correctly recalled using a neurally feasible algorithm.

## 1 Introduction

Neural associative memories with exponential storage capacity and large (potentially linear) fraction of error-correction guarantee have been the topic of extensive research for the past three decades. A networked associative memory model must have the ability to learn and remember an arbitrary but specific set of $n$-length messages. At the same time, when presented with a noisy query, i.e., an $n$-length vector close to one of the messages, the system must be able to recall the correct message. While the first task is called the *learning phase*, the second one is referred to as the *recall phase*.

Associative memories are traditionally modeled by what is called binary Hopfield networks [15], where a weighted graph of size $n$ is considered with each vertex representing a binary state neuron. The edge-weights of the network are learned from the set of binary vectors to be stored by the Hebbian learning rule [13]. It has been shown in [22] that, to recover the correct vector in the presence of a linear (in $n$) number of errors, it is not possible to store more than $O(\frac{n}{\log n})$ arbitrary binary vectors in the above model of learning. In the pursuit of networks that can store exponential (in $n$) number of messages, some works [26, 12, 21] do show the existence of Hopfield networks that can store $\sim 1.22^n$ messages. However, for such Hopfield networks, only a small number of errors in the query render the recall phase unsuccessful. The Hopfield networks that store non-binary message vectors are studied in [17, 23], where the storage capacity of such networks against large fraction of errors is again shown to be linear in $n$. There have been multiple efforts to increase the storage capacity of the associative memories to exponential by moving away from the framework of the Hopfield networks (in term of both the learning and the recall phases)[14, 11, 19, 25, 18]. These efforts also involve relaxing the requirement of storing the collections of arbitrary messages. In [11], Gripon and Berrou stored $O(n^2)$ number of sparse message vectors in the form of neural cliques. Another setting where neurons have been assumed to have a large (albeit constant) number

---

[*]This work was done when the author was with the Dept. of ECE, University of Texas at Austin, TX, USA.

of states, and at the same time the message set (or the *dataset*) is assumed to form a linear subspace is considered in [19, 25, 18].

The most basic premise of the works on neural associative memory is to design a graph dynamic system such that the vectors to be stored are the steady states of the system. One way to attain this is to learn a set of constraints that every vector in the dataset must satisfy. The inclusion relation between the variables in the vectors and the constraints can be represented by a bipartite graph (cf. Fig. 1). For the recall phase, noise removal can be done by running belief propagation on this bipartite graph. It can be shown that the correct message is recovered successfully under conditions such as sparsity and expansion properties of the graph. This is the main idea that has been explored in [19, 25, 18]. In particular, under the assumption that the messages belong to a linear subspace, [19, 25] propose associative memories that can store exponential number of messages while tolerating at most constant number of errors. This approach is further refined in [18], where each message vector from the dataset is assumed to comprise overlapping sub-vectors which belong to different linear subspaces. The learning phase finds the (sparse) linear constraints for the subspaces as-

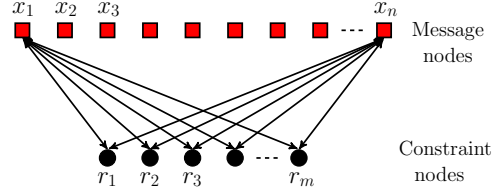

Figure 1: The complete bipartite graph corresponding to the associative memory. Here, we depict only a small fraction of edges. The edge weights of the bipartite graph are obtained from the linear constraints satisfied by the messages. Information can flow in both directions in the graph, i.e., from a message node to a constraint node and from a constraint node to a message node. In the steady state $n$ message nodes store $n$ coordinates of a valid message, and all the $m$ constraints nodes are satisfied, i.e., the weighted sum of the values stored on the neighboring message nodes (according to the associated edge weights) is equal to zero. Note that an edge is relevant for the information flow iff the corresponding edge weight is nonzero.

sociated with these sub-vectors. For the recall phase then belief propagation decoding ideas of error-correcting codes have been used. In [18], Karbasi et al. show that the associative memories obtained in this manner can store exponential (in $n$) messages. They further show that the recall phase can correct linear (in $n$) number of random errors provided that the bipartite graph associated with learnt linear constraints (during learning phase) has certain structural properties.

Our work is very closely related to the above principle. Instead of finding a sparse set of constraints, we aim to find a set of linear constraints that satisfy 1) the coherence property, 2) the null-space property or 3) the restricted isometry property (RIP). Indeed, for a large class of random signal models, we show that, such constraints must exists and can be found in polynomial time. Any of the three above mentioned properties provide sufficient condition for recovery of sparse signals or vectors [8, 6]. Under the assumption that the noise in the query vector is sparse, denoising can be done very efficiently via iterative sparse recovery algorithms that are *neurally feasible* [9]. A neurally feasible algorithm for our model employs only local computations at the vertices of the corresponding bipartite graph based on the information obtained from their neighboring nodes.

## 1.1 Our techniques and results

Our main provable results pertain to two different models of datasets, and are given below.

**Theorem 1** (Associative memory with sub-gaussian dataset model)**.** *It is possible to store a dataset of size $\sim \exp(n^{3/4})$ of $n$-length vectors in a neural network of size $O(n)$ such that a neurally feasible algorithm can output the correct vector from the dataset given a noisy version of the vector corrupted in $\Theta(n^{1/4})$ coordinates.*

**Theorem 2** (Associative memory with dataset spanned by random rows of fixed orthonormal basis)**.** *It is possible to store a dataset of size $\sim \exp(r)$ of $n$-length vectors in a neural network of size $O(n)$ such that a neurally feasible algorithm can output the correct vector from the dataset given a noisy version of the vector corrupted in $\Theta(\frac{n-r}{\log^6 n})$ coordinates.*

Theorem 1 follows from Prop. 1 and Theorem 3, while Theorem 2 follows from Prop. 2 and 3; and by also noting the fact that all $r$-vectors over any finite alphabet can be linearly mapped to $\exp(r)$ number of points in a space of dimensionality $r$. The neural feasibility of the recovery follows from the discussion of Sec. 5. In contrast with [18], our sparse recovery based approach provides

associative memories that are robust against stronger error model which comprises adversarial error patterns as opposed to random error patterns. Even though we demonstrate the associative memories which have sub-exponential storage capacity and can tolerate sub-linear (but polynomial) number of errors, neurally feasible recall phase is guaranteed to recover the message vector from adversarial errors. On the other hand, the recovery guarantees in [18, Theorem 3 and 5] hold if the bipartite graph obtained during learning phase possesses certain structures (e.g. degree sequence). However, it is not apparent in their work if the learnt bipartite graph indeed has these structural properties. Similar to the aforementioned papers, our operations are performed over real numbers. We show the dimensionality of the dataset to be large enough, as referenced in Theorem 1 and 2. As in previous works such as [18], we can therefore find a large number of points, exponential in the dimensionality, with finite (integer) alphabet that can be treated as the message vectors or dataset.

Our main contribution is to bring in the model of sparse recovery in the domain of associative memory - a very natural connection. The main techniques that we employ are as follows: 1) In Sec. 3, we present two models of ensembles for the dataset. The dataset belongs to subspaces that have associated orthogonal subspace with 'good' basis. These good basis for the orthogonal subspaces satisfy one or more of the conditions introduced in Sec. 2, a section that provides some background material on sparse recovery and various sufficient conditions relevant to the problem. 2) In Sec. 4, we briefly describe a way to obtain a 'good' null basis for the dataset. The found bases serve as measurement matrices that allow for sparse recovery. 3) Sec. 5 focus on the recall phases of the proposed associative memories. The algorithms are for sparse recovery, but stated in a way that are implementable in a neural network.

In Sec. 6, we present some experimental results showcasing the performance of the proposed associative memories. In Appendix C, we describe another approach to construct associative memory based on the dictionary learning problem [24].

## 2 Definition and mathematical preliminaries

**Notation:** We use lowercase boldface letters to denote vectors. Uppercase boldface letters represent matrices. For a matrix $\mathbf{B}$, $\mathbf{B}^T$ denotes the transpose of $\mathbf{B}$. A vector is called $k$-sparse if it has only $k$ nonzero entries. For a vector $\mathbf{x} \in \mathbb{R}^n$ and any set of coordinates $I \subseteq [n] \equiv \{1, 2, \ldots, n\}$, $\mathbf{x}_I$ denotes the projection of $\mathbf{x}$ on to the coordinates of $I$. For any set of coordinates $I \subseteq [n]$, $I^c \equiv [n] \setminus I$. Similarly, for a matrix $B$, $B_I$ denotes the sub-matrix obtained by the rows of $\mathbf{B}$ that are indexed by the set $I$. We use $\mathrm{span}(\mathbf{B})$ to denote the subspace spanned by the columns of $\mathbf{B}$. Given an $m \times n$ matrix $\mathbf{B}$, denote the columns of the matrix as $\mathbf{b}_j, j = 1, \ldots, n$ and assume, for all the matrices in this section, that the columns are all unit norm, i.e., $\|\mathbf{b}_j\|^2 = 1$.

**Definition 1** (Coherence)**.** The *mutual coherence* of the matrix $\mathbf{B}$ is defined to be

$$\mu(\mathbf{B}) = \max_{i \neq j} |\langle \mathbf{b}_i, \mathbf{b}_j \rangle|. \tag{1}$$

**Definition 2** (Null-space property)**.** The matrix $\mathbf{B}$ is supposed to satisfy the *null-space property* with parameters $(k, \alpha < 1)$ if $\|\mathbf{h}_I\|_1 \leq \alpha \|\mathbf{h}_{I^c}\|_1$, for every vector $\mathbf{h} \in \mathbb{R}^n$ with $\mathbf{Bh} = 0$ and any set $I \subseteq [n], |I| = k$.

**Definition 3** (RIP)**.** A matrix $\mathbf{B}$ is said to satisfy the restricted isometry property with parameters $k$ and $\delta$, or the the $(k, \delta)$-RIP, if for all $k$-sparse vectors $\mathbf{x} \in \mathbb{R}^n$,

$$(1 - \delta)\|\mathbf{x}\|_2^2 \leq \|\mathbf{Bx}\|_2^2 \leq (1 + \delta)\|\mathbf{x}\|_2^2. \tag{2}$$

Next we list some results pertaining to sparse signal recovery guarantee based on these aforementioned parameters. The sparse recovery problem seeks the solution $\hat{\mathbf{x}}$, that has the smallest number of nonzero entries, of the underdetermined system of equation $\mathbf{Bx} = \mathbf{r}$, where, $\mathbf{B} \in \mathbb{R}^{m \times n}$ and $\mathbf{x} \in \mathbb{R}^n$. The *basis pursuit* algorithm for sparse recovery provides the following estimate.

$$\hat{\mathbf{x}} = \arg \min_{\mathbf{Bx} = \mathbf{r}} \|\mathbf{x}\|_1. \tag{3}$$

Let $\mathbf{x}_k$ denote the projection of $\mathbf{x}$ on its largest $k$ coordinates.

**Proposition 1.** *If* $\mathbf{B}$ *has* null-space property *with parameters* $(k, \alpha < 1)$*, then, we have,*

$$\|\hat{\mathbf{x}} - \mathbf{x}\|_1 \leq \frac{2(1 + \alpha)}{1 - \alpha} \|\mathbf{x} - \mathbf{x}_k\|_1. \tag{4}$$

The proof of this is quite standard and delegated to the Appendix A.

**Proposition 2** ([5]). *The $(2k, \sqrt{2} - 1)$-RIP of the sampling matrix implies, for a constant $c$,*

$$\|\hat{\mathbf{x}} - \mathbf{x}\|_2 \leq \frac{c}{\sqrt{k}}\|\mathbf{x} - \mathbf{x}_k\|_1. \tag{5}$$

Furthermore, it can be easily seen that any matrix is $(k, (k - 1)\mu)$-RIP, where $\mu$ is the mutual coherence of the sampling matrix.

## 3 Properties of the datasets

In this section, we show that, under reasonable random models that represent quite general assumptions on the datasets, it is possible to learn linear constraints on the messages, that satisfy one of the sufficient properties of sparse recovery: incoherence, null-space property or RIP. We mainly consider two models for the dataset: 1) sub-gaussian model 2) span of a random set from an orthonormal basis.

### 3.1 Sub-Gaussian model for the dataset and the null-space property

In this section we consider the message sets that are spanned by a basis matrix which has its entries distributed according to a sub-gaussain distribution. The sub-gaussian distributions are prevalent in machine learning literature and provide a broad class of random models to analyze and validate various learning algorithms. We refer the readers to [27, 10] for the background on these distribution. Let $\mathbf{A} \in \mathbb{R}^{n \times r}$ be an $n \times r$ random matrix that has independent zero mean sub-gaussian random variables as its entries. We assume that the subspace spanned by the columns of the matrix $\mathbf{A}$ represents our dataset $\mathcal{M}$. The main result of this section is the following.

**Theorem 3.** *The dataset above satisfies a set of linear constraints, that has the null-space property. That is, for any $\mathbf{h} \in \mathcal{M} \equiv \mathrm{span}(\mathbf{A})$, the following holds with high probability:*

$$\|\mathbf{h}_I\|_1 \leq \alpha\|\mathbf{h}_{I^c}\|_1 \quad \text{for all } I \subseteq [n] \text{ such that } |I| \leq k, \tag{6}$$

*for $k = O(n^{1/4})$, $r = O(n/k)$ and a constant $\alpha < 1$.*

The rest of this section is dedicated to the proof of this theorem. But, before we present the proof, we state a result from [27] which we utilize to prove Theorem 3.

**Proposition 3.** *[27, Theorem 5.39] Let $\mathbf{A}$ be an $s \times r$ matrix whose rows $\mathbf{a}_i$ are independent sub-gaussian isotropic random vectors in $\mathbb{R}^n$. Then for every $t \geq 0$, with probability at least $1 - 2\exp(-ct^2)$ one has*

$$\sqrt{s} - C\sqrt{r} - t \leq s_{\min}(\mathbf{A}) = \min_{\mathbf{x} \in \mathbb{R}^r : \|\mathbf{x}\|_2 = 1} \|\mathbf{A}\mathbf{x}\|_2$$

$$\leq s_{\max}(\mathbf{A}) = \max_{\mathbf{x} \in \mathbb{R}^r : \|\mathbf{x}\|_2 = 1} \|\mathbf{A}\mathbf{x}\|_2 \leq \sqrt{s} + C\sqrt{r} + t. \tag{7}$$

*Here $C$ and $c$ depends on the sub-gaussian norms of the rows of the matrix $\mathbf{A}$.*

*Proof of Theorem 3:* Consider an $n \times r$ matrix $\mathbf{A}$ which has independent sub-gaussian isotropic random vectors as its rows. Now for a given set $I \subseteq [n]$, we can focus on two disjoint sub-matrices of $\mathbf{A}$: 1) $\mathbf{A}_1 = \mathbf{A}_I$ and 2) $\mathbf{A}_2 = \mathbf{A}_{I^c}$.

Using Proposition 3 with $s = |I|$, we know that, with probability at least $1 - 2\exp(-ct^2)$, we have

$$s_{\max}(\mathbf{A}_1) = \max_{\mathbf{x} \in \mathbb{R}^r : \|\mathbf{x}\|_2 = 1} \|\mathbf{A}_1\mathbf{x}\|_2 \leq \sqrt{|I|} + C\sqrt{r} + t. \tag{8}$$

Since we know that $\|\mathbf{A}_1\mathbf{x}\|_1 \leq \sqrt{|I|}\|\mathbf{A}_1\mathbf{x}\|_2$, using (8) the following holds with probability at least $1 - 2\exp(-ct^2)$.

$$\|(\mathbf{A}\mathbf{x})_I\|_1 = \|\mathbf{A}_1\mathbf{x}\|_1 \leq \sqrt{|I|}\|\mathbf{A}_1\mathbf{x}\|_2 \leq |I| + C\sqrt{|I|r} + t\sqrt{|I|} \quad \forall \mathbf{x} \in \mathbb{R}^r : \|\mathbf{x}\|_2 = 1. \tag{9}$$

We now consider $\mathbf{A}_2$. It follows from Proposition 3 with $s = |I^c| = n - |I|$ that with probability at least $1 - 2\exp(-ct^2)$,

$$s_{\min}(\mathbf{A}_2) = \min_{\mathbf{x} \in \mathbb{R}^r : \|\mathbf{x}\|_2 = 1} \|\mathbf{A}_2 \mathbf{x}\|_2 \geq \sqrt{n - |I|} - C\sqrt{r} - t. \tag{10}$$

Combining (10) with the observation that $\|\mathbf{A}_2\mathbf{x}\|_1 \geq \|\mathbf{A}_2\mathbf{x}\|_2$, the following holds with probability at least $1 - 2\exp(-ct^2)$.

$$\|(\mathbf{A}\mathbf{x})_{I^c}\|_1 = \|\mathbf{A}_2\mathbf{x}\|_1 \geq \|\mathbf{A}_2\mathbf{x}\|_2 \geq \sqrt{n - |I|} - C\sqrt{r} - t \quad \text{for all } \mathbf{x} \in \mathbb{R}^r : \|\mathbf{x}\|_2 = 1. \tag{11}$$

Note that we are interested in showing that for all $\mathbf{h} \in \mathcal{M}$, we have

$$\|\mathbf{h}_I\|_1 \leq \alpha \|\mathbf{h}_{I^c}\|_1 \quad \text{for all } I \subseteq [n] \text{ such that } |I| \leq k. \tag{12}$$

This is equivalent to showing that the following holds for all $\mathbf{x} \in \mathbb{R}^r : \|\mathbf{x}\|_2 = 1$.

$$\|(\mathbf{A}\mathbf{x})_I\|_1 \leq \alpha \|(\mathbf{A}\mathbf{x})_{I^c}\|_1 \quad \text{for all } I \subseteq [n] \text{ such that } |I| \leq k. \tag{13}$$

For a given $I \subseteq [n]$, we utilize (9) and (11) to guarantee that (13) holds with probability at least $1 - 2\exp(-ct^2)$ as long as

$$|I| + C\sqrt{|I|r} + t\sqrt{|I|} \leq \alpha\left(\sqrt{n - |I|} - C\sqrt{r} - t\right) \tag{14}$$

Now, given that $k = |I|$ satisfies (14), (13) holds for all $I \subset [n] : |I| = k$ with probability at least

$$1 - 2\binom{n}{k}\exp(-ct^2) \geq 1 - \left(\frac{en}{k}\right)^k \exp(-ct^2). \tag{15}$$

Let's consider the following set of parameters: $k = O(n^{1/4})$, $r = O(n/k) = O(n^{3/4})$ and $t = \Theta(\sqrt{k\log(n/k)})$. This set of parameters ensures that (14) holds with overwhelming probability (cf. (15)). $\qquad\square$

**Remark 1.** *In Theorem 3, we specify one particular set of parameters for which the null-space property holds. Using (14) and (15), it can be shown that the null-space property in general holds for the following set of parameters: $k = O(\sqrt{n}/\log n)$, $r = O(n/k)$ and $t = \Theta(\sqrt{k\log(n/k)})$. Therefore, it possible to trade-off the number of correctable errors during the recall phase (denoted by $k$) with the dimension of the dataset (represented by $r$).*

### 3.2 Span of a random set of columns of an orthonormal basis

Next, in this section, we consider the ensemble of signals spanned by a random subset of rows from a fixed orthonormal basis $\mathbf{B}$. Assume $\mathbf{B}$ to be an $n \times n$ matrix with orthonormal rows. Let $\Gamma \subset [n]$ be a random index set such that $\mathbb{E}(|\Gamma|) = r$. The vectors in the dataset have form $\mathbf{h} = \mathbf{B}_\Gamma^T \mathbf{u}$ for some $\mathbf{u} \in \mathbb{R}^{|\Gamma|}$. In other words, the dataset $\mathcal{M} \equiv \text{span}(\mathbf{B}_\Gamma^T)$.

In this case, $\mathbf{B}_{\Gamma^c}$ constitutes a basis matrix for the null space of the dataset. Since we have selected the set $\Gamma$ randomly, set $\Omega \equiv \Gamma^c$ is also a random set with $\mathbb{E}(\Omega) = n - \mathbb{E}(\Gamma) = n - r$.

**Proposition 4** ([7]). *Assume that $\mathbf{B}$ be an $n \times n$ orthonormal basis for $\mathbb{R}^n$ with the property that $\max_{i,j} |\mathbf{B}_{i,j}| \leq \nu$. Consider a random $|\Omega| \times n$ matrix $\mathbf{C}$ obtained by selecting a random set of rows of $\mathbf{B}$ indexed by the set $\Omega \in [n]$ such that $\mathbb{E}(\Omega) = m$. Then the matrix $\mathbf{C}$ obeys $(k, \delta)$-RIP with probability at least $1 - O(n^{-\rho/\alpha})$ for some fixed constant $\rho > 0$, where $k = \frac{\alpha m}{\nu^2 \log^6 n}$.*

Therefore, we can invoke Proposition 4 to conclude that the matrix $\mathbf{B}_{\Gamma^c}$ obeys $(k, \delta)$-RIP with $k = \frac{\alpha(n-r)}{\nu^2 \log^6 n}$ with $\nu$ being the largest absolute value among the entries of $\mathbf{B}_{\Gamma^c}$.

## 4 Learning the constraints: null space with small coherence

In the previous section, we described some random ensemble of datasets that can be stored on an associative memory based on sparse recovery. This approach involves finding a basis for the

orthogonal subspace to the message or the signal subspace (dataset). Indeed, our learning algorithm simply finds a null space of the dataset $\mathcal{M}$. While obtaining the basis vectors of $\text{null}(\mathcal{M})$, we require them to satisfy null-space property, RIP or small mutual coherence so that the a signal can be recovered from its noisy version via the basis pursuit algorithm, that can be neurally implemented (see Sec. 5.2). However, for a given set of message vectors, it is computationally intractable to check if the obtained (learnt) orthogonal basis has null-space property or RIP with suitable parameters associated with these properties. Mutual coherence of the orthogonal basis, on the other hand, can indeed be verified in a tractable manner. Further, the more straight-forward iterative soft thresholding algorithm will be successful if $\text{null}(\mathcal{M})$ has low coherence. This will also lead to fast convergence of the recovery algorithm (see, Sec. 5.1). Towards this, we describe one approach that ensures the selection of a orthogonal basis that has smallest possible mutual coherence. Subsequently, using the mutual coherence based recovery guarantees for sparse recovery, this basis enables an efficient recovery phase for the associative memory.

One underlying assumption on the dataset that we make is its less than full dimensionality. That is, the dataset must belong to a low dimensional subspace, so that its null-space is not trivial. In practical cases, $\mathcal{M}$ is approximately low-dimensional. We use a preprocessing step, employing *principal component analysis* (PCA) to make sure that the dataset is low-dimensional. We do not indulge in to a more detailed description of this phase, as it seems to be quite standard (see, [18]).

---

**Algorithm 1** Find null-space with low coherence

---

**Input:** The dataset $\mathcal{M}$ with $n$ dimensional vectors. An initial coherence $\mu_0$ and a step-size $\lambda$
**Output:** A $m \times n$ orthogonal matrix $\mathbf{B}$ and coherence $\mu$
**Preprocessing.** Perform PCA on $\mathcal{M}$
Find the $n \times r$ basis matrix $\mathbf{A}$ of $\mathcal{M}$
**for** $l = 0, 1, 2, \ldots$ **do**
   Find a feasible point of the quadratically constrained quadratic problem (QCQP) below (interior point method): $\mathbf{BA} = 0$; $\|\mathbf{b}_i\| = 1, \forall i \in [n]$; $|\langle \mathbf{b}_i, \mathbf{b}_j \rangle| \leq \mu_l$ where $\mathbf{B}$ is $(n-r) \times n$
   **if** No feasible point found **then**
     break
   **else**
     $\mu \leftarrow \mu_l$
     $\mu_{l+1} = \mu_l - \lambda$
   **end if**
**end for**

---

## 5  Recall via neurally feasible algorithms

We now focus on the second aspect of an associative memory, namely the recovery phase. For the signal model that we consider in this paper, the recovery phase is equivalent to solving a sparse signal recovery problem. Given a noisy vector $\mathbf{y} = \mathbf{x} + \mathbf{e}$ from the dataset, we can use the basis of the null-space $\mathbf{B}$ associated to our dataset that we constructed during the learning phase to obtain $\mathbf{r} = \mathbf{By} = \mathbf{Be}$. Now given that $\mathbf{e}$ is sufficiently sparse enough and the matrix $\mathbf{B}$ obeys the properties of Sec. 2, we can solve for $\mathbf{e}$ using a sparse recovery algorithm. Subsequently, we can remove the error vector $\mathbf{e}$ from the noisy signal $\mathbf{y}$ to construct the underlying message vector $\mathbf{x}$. There is a plethora of algorithms available in the literature to solve this problem. However, we note that for the purpose of an associative memory, the recovery phase should be neurally feasible and computationally simple. In other words, each node (or storage unit) should be able to recover the coordinated associated to it locally by applying simple computation on the information received from its neighboring nodes (potentially in an iterative manner).

### 5.1  Recovery via Iterative soft thresholding (IST) algorithm

Among the various, sparse recovery algorithms in the literature, iterative soft thresholding (IST) algorithm is a natural candidate for implementing the recovery phase of the associative memories with underlying setting. The IST algorithm tries to solve the following unconstrained $\ell_1$-regularized

least square problem which is closely related to the basis pursuit problem described in (3) and (18).

$$\hat{\mathbf{e}} = \arg\min_{\mathbf{e}} \nu \|\mathbf{e}\|_1 + \frac{1}{2}\|\mathbf{B}\mathbf{e} - \mathbf{r}\|^2. \tag{16}$$

For the IST algorithm, its $t + 1$-th iteration is described as follows.

$$\text{(IST)} \qquad \mathbf{e}^{t+1} = \eta^S(\mathbf{e}^t - \tau\mathbf{B}^T(\mathbf{B}\mathbf{e}^t - \mathbf{r}); \lambda = \tau\nu). \tag{17}$$

Here, $\tau$ is a constant and $\eta^S(\mathbf{x}; \lambda) = (\text{sgn}(x_1)(x_1 - \lambda)^+, \text{sgn}(x_2)(x_2 - \lambda)^+, \dots, \text{sgn}(x_n)(x_n - \lambda)^+)$ denotes the soft thresholding (or shrinkage) operator. Note that the IST algorithm as described in (17) is neurally feasible as it involves only 1) performing matrix vector multiplications and 2) soft thresholding a coordinate in a vector independent of the values of other coordinates in the vector. In Appendix B, we describe in details how the IST algorithm can be performed over a bipartite neural network with $\mathbf{B}$ as its edge weight matrix. Under suitable assumption on the coherence of the measurement matrix $\mathbf{B}$, the IST algorithm is also known to converge to the correct $k$-sparse vector $\mathbf{e}$ [20]. In particular, Maleki [20] allows the thresholding parameter $\lambda$ to be varied in every iteration such that all but at most the largest $k$ coordinates (in terms of their absolute values) are mapped to zero by the soft thresholding operation. In this setting, Maleki shows that the solution of the IST algorithm recovers the correct support of the optimal solution in finite steps and subsequently converges to the true solution very fast. However, we are interested in analysis of the IST algorithm in a setting where thresholding parameter is kept a suitable constant depending on other system parameters so that the algorithm remains neurally feasible. Towards this, we note that there exists general analysis of the IST algorithm even without the coherence assumption.

**Proposition 5.** *[4, Theorem 3.1] Let $\{\mathbf{e}^t\}_{t\geq 1}$ be as defined in (17) with $\frac{1}{\tau} \geq \lambda_{\max}(\mathbf{B}^T\mathbf{B})$[1]. Then, for any $t \geq 1$, $h(\mathbf{e}^t) - h(\mathbf{e}) \leq \frac{\lambda\|\mathbf{e}^0 - \mathbf{e}\|^2}{2t}$. Here, $h(\mathbf{e}) = \frac{1}{2}\|\mathbf{r} - \mathbf{B}\mathbf{e}\|^2 + \nu\|\mathbf{e}\|_1$ is the objective function defined in (16).*

## 5.2 Recovery via Bregman iterative algorithm

Recall that the basis pursuit algorithm refers to the following optimization problem.

$$\hat{\mathbf{e}} = \arg\min_{\mathbf{e}} \{\|\mathbf{e}\|_1 \ : \ \mathbf{r} = \mathbf{B}\mathbf{e}\}. \tag{18}$$

Even though the IST algorithm as described in the previous subsection solves the problem in (16), the parameter value $\nu$ needs to be set small enough so that the recovered solution $\hat{\mathbf{e}}$ nearly satisfies the constraint $\mathbf{B}\mathbf{e} = \mathbf{r}$ in (18). However, if we insist on recovering the solution $\mathbf{e}$ which exactly meets the constraint, one can employ the Bregman iterative algorithm from [29]. The Bregman iterative algorithm relies on the Bregman distance $D^{\mathbf{p}}_{\|\cdot\|_1}(\cdot, \cdot)$ based on $\|\cdot\|_1$ which is defined as follows.

$$D^{\mathbf{p}}_{\|\cdot\|_1}(\mathbf{e}_1, \mathbf{e}_2) = \|\mathbf{e}_1\|_1 - \|\mathbf{e}_2\|_1 - \langle\mathbf{p}, \mathbf{e}_1 - \mathbf{e}_2\rangle,$$

where $\mathbf{p} \in \partial\|\mathbf{e}_2\|_1$ is a sub-gradient of the $\ell_1$-norm at the point $\mathbf{e}_2$. The $(t + 1)$-th iteration of the Bregman iterative algorithm is then defined as follows.

$$\mathbf{e}^{t+1} = \arg\min_{\mathbf{e}} D^{\mathbf{p}^t}_{\|\cdot\|_1}(\mathbf{e}, \mathbf{e}^t) + \frac{1}{2}\|\mathbf{B}\mathbf{e} - \mathbf{r}\|^2,$$

$$= \arg\min_{\mathbf{e}} \|\mathbf{e}\|_1 - (\mathbf{p}^t)^T\mathbf{e} + \frac{1}{2}\|\mathbf{B}\mathbf{e} - \mathbf{r}\|^2 - \|\mathbf{e}^t\|_1 + (\mathbf{p}^t)^T\mathbf{e}^t, \tag{19}$$

$$\mathbf{p}^{t+1} = \mathbf{p}^t - \mathbf{B}^T(\mathbf{B}\mathbf{e}^{t+1} - \mathbf{r}). \tag{20}$$

Note that, for the $(t + 1)$-th iteration, the objective function in (19) is essentially equivalent to the objective function in (16). Therefore, each iteration of the Bergman iterative algorithm can be solved using the IST algorithm. It is shown in [29] that after a finite number of iteration of the Bregman iterative algorithm, one recovers the solution of the problem in (18) (Theorem 3.2 and 3.3 in [29]).

**Remark 2.** *We know that the IST algorithm is neurally feasible. Furthermore, the step described in (20) is neurally feasible as it only involve matrix-vector multiplications in the spirit of Eq. (17). Since each iteration of the Bregman iterative algorithm only relies on these two operations, it follows that the Bregman iterative algorithm is neurally feasible as well. It should be noted that the neural feasibility of the Bregman iterative algorithm was discussed in [16] as well, however the neural structures employed by [16] is different from ours.*

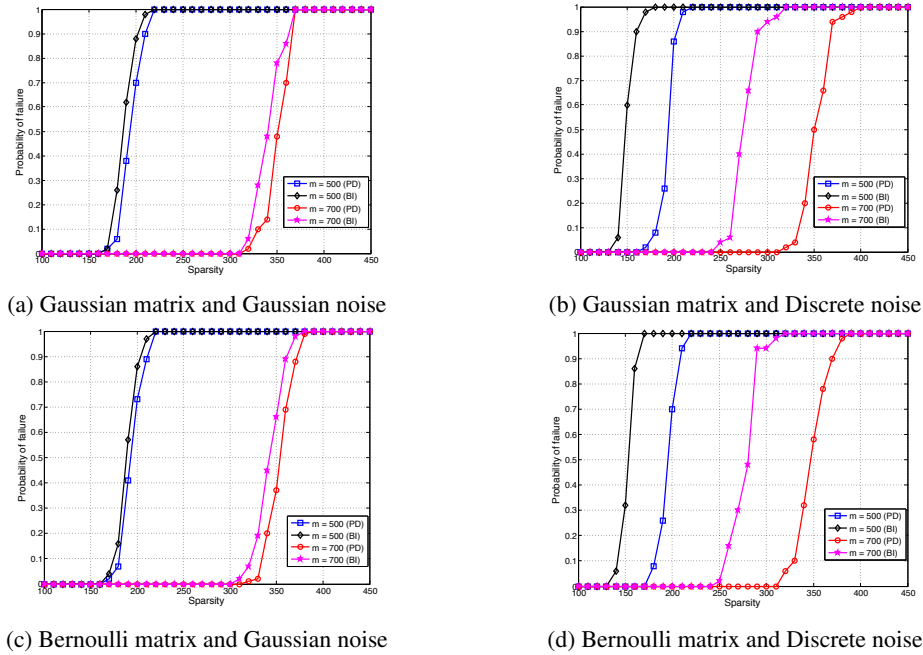

| | |
|---|---|
| (a) Gaussian matrix and Gaussian noise | (b) Gaussian matrix and Discrete noise |
| (c) Bernoulli matrix and Gaussian noise | (d) Bernoulli matrix and Discrete noise |

Figure 2: Performance of the proposed associative memory approach during recall phase. The PD algorithm refers to the primal dual algorithm to solve linear program associated with the problem in (18). The BI algorithm refers to the Bregman iterative algorithm described in Sec. 5.2.

# 6 Experimental results

In this section, we demonstrate the feasibility of the associative memory framework using computer generated data. Along the line of the discussion in Sec. 3.1, we first sample an $n \times r$ sub-gaussian matrix $\mathbf{A}$ with i.i.d entries. We consider two sub-gaussian distributions: 1) Gaussian distribution and 2) Bernoulli distribution over $\{+1, -1\}$. The message vectors to be stored are then assumed to be spanned by the $k$ columns of the sampled matrix. For the learning phase, we find a good basis for the subspace orthogonal to the space spanned by the columns of the matrix $\mathbf{A}$. For noise during the recall phase, we consider two noise models: 1) Gaussian noise and 2) discrete noise where each nonzero elements take value in the set $\{-M, -(M-1), \ldots, M\}\backslash\{0\}$.

Figure 2 presents our simulation results for $n = 1000$. For recall phase, we employ the Bregman iterative (BI) algorithm with the IST algorithm as a subroutine. We also plot the performance of the primal dual (PD) algorithm based linear programming solution for the recovery problem of interest (cf. (18)). This allows us to gauge the disadvantage due to the restriction of working with a neurally feasible recovery algorithm, e.g., the BI algorithm in our case. Furthermore, we consider message sets with two different dimensions which amounts to $m = 500$ and $m = 700$. Note that the dimension of the message set is $n - m$. We run 50 iterations of the recovery algorithms for a given set of parameters to obtain the estimates of the probability of failure (of exact recovery of error vector). In Fig. 2a, we focus on the setting with Gaussian basis matrix (for message set) and unit variance zero mean Gaussian noise during the recall phase. It is evident that the proposed associative memory do allow for the exact recovery of error vectors up to certain sparsity level. This corroborate our findings in Sec. 3. We also note that the performance of the BI algorithm is very close to the PD algorithm. Fig. 2b shows the performance of the recall phase for the setting with Gaussian basis for message set and discrete noise model with $M = 4$. In this case, even though the BI algorithm is able to exactly recover the noise vector up to a particular sparsity level, it's performance is worse than that of PD algorithm. The performance of the recall phase with Bernoulli bases matrices for message set is shown in Fig. 2c and 2d. The results are similar to those in the case of Gaussian bases matrices for the message sets.

## Footnotes

[1]Note that $\lambda_{\max}(\mathbf{B}^T\mathbf{B})$, the maximum eigenvalue of the matrix $\mathbf{B}^T\mathbf{B}$ serves as a Lipschitz constant for the gradient $\delta f(\mathbf{e})$ of the function $f(\mathbf{e}) = \frac{1}{2}\|\mathbf{r} - \mathbf{B}\mathbf{e}\|^2$

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
