[Supplementary Material]



Figure 3: At the beginning of the recall phase, the associative memory is presented with a noisy version of a message $\mathbf{y} = \mathbf{x} + \mathbf{e}$. The $j$-th message node stores $y_j$. The $m$ constraint nodes take values according to the vector $\mathbf{r} = \mathbf{B}\mathbf{y} = \mathbf{B}\mathbf{e}$. Note that the $m \times n$ matrix $\mathbf{B}$ denotes the edge weight of the bipartite graph corresponding to the associative memory.

## A Proof of Prop. 1

Assume, $\mathbf{h} = \hat{\mathbf{x}} - \mathbf{x}$, then

$$
\begin{aligned}
\|\mathbf{x}\|_1 \geq \|\hat{\mathbf{x}}\|_1 = \|\mathbf{x} + \mathbf{h}\|_1 &= \|\mathbf{x}_T + \mathbf{h}_T\|_1 + \|\mathbf{x}_{T^c} + \mathbf{h}_{T^c}\|_1 \\
&\geq \|\mathbf{x}_T\|_1 - \|\mathbf{h}_T\|_1 - \|\mathbf{x}_{T^c}\|_1 + \|\mathbf{h}_{T^c}\|_1
\end{aligned}
$$

$$
\implies \|\mathbf{h}_{T^c}\|_1 \leq \|\mathbf{h}_T\|_1 + 2\|\mathbf{x}_{T^c}\|_1,
$$

where $T \in \{1, \ldots, n\}$ is the largest $k$ coordinates of $\mathbf{x}$ and $T^c = \{1, \ldots, n\} \setminus T$. If $\mathbf{B}$ has *null-space property* with parameters $(k, \alpha)$, then, we have,

$$
\|\mathbf{h}_{T^c}\|_1 \leq 2\|\mathbf{x}_{T^c}\|_1 + \alpha\|\mathbf{h}_{T^c}\|_1.
$$

Since, $\|\mathbf{h}\|_1 \leq (1 + \alpha)\|\mathbf{h}_{T^c}\|_1$,

$$
\frac{1 - \alpha}{1 + \alpha}\|\mathbf{h}\|_1 \leq 2\|\mathbf{x}_{T^c}\|_1.
$$

Hence the null-space property of sampling matrix implies,

$$
\|\hat{\mathbf{x}} - \mathbf{x}\|_1 \leq \frac{2(1 + \alpha)}{1 - \alpha}\|\mathbf{x} - \mathbf{x}_k\|_1. \tag{21}
$$

## B Neural feasibility of recall phase

In this section, we briefly comment on the neural feasibility of the recovery algorithms presented in Sec. 5. See Fig. 3 for the structure of the associative memory. The edge weights of the bipartite graph are defined by the $m \times n$ matrix $\mathbf{B}$ which is a basis matrix of the subspace orthogonal to the message set. Thus, the edge weight between $i$-th constraint node and $j$-th message node in $B_{i,j}$. Given access to a noisy observation $\mathbf{y} = \mathbf{x} + \mathbf{e}$, the $j$-th message node store $y_j$ and the $i$-th constraint node stores $r_i$, where $\mathbf{r} = \mathbf{B}\mathbf{e}$. The objective of the recall phase is to enable recovery of $e_j$ at the $j$-th message node. Towards, this the message nodes start computation with an initial estimate $\mathbf{e}^0 = 0$ for the vector $\mathbf{e}$. Here, we describe how the IST algorithm as defined in (17) can be implemented over the bipartite graph. Since the Bregman iterative algorithm involves similar steps in its iterations, its neural feasibility also follows from that of the IST algorithm.

Note that the $(t + 1)$-th iteration of the IST algorithm is defined as follows.

$$
\mathbf{e}^{t+1} = \eta^S(\mathbf{e}^t - \tau\mathbf{B}^T(\mathbf{B}\mathbf{e}^t - \mathbf{r}); \lambda = \tau\nu).
$$

This can be divided into three basic sequential computation tasks: 1) computation of $\mathbf{B}\mathbf{e}^t - \mathbf{r}$ at the constraint nodes, 2) computation of $\mathbf{e}^t - \tau\mathbf{B}^T(\mathbf{B}\mathbf{e}^t - \mathbf{r})$ at the message nodes, and 3) coordinate-wise soft thresholding operation $\eta^S(\cdot, \tau\nu)$ at the message nodes. Assuming that the message nodes store $\mathbf{e}^t$ from the computation from the previous iteration, the $i$-th constraint node can now compute

$$
\tilde{r}_i^{t+1} \leftarrow \sum_{j=1}^{n} B_{i,j}e_j^t - r_j.
$$

This completes the first task. Note that each constraint node only accesses the weighted information stored on its neighbor during the computation. For second task, based on the information collected from its neighboring constraints nodes, the $j$-th message node now computes

$$\tilde{e}_j^{t+1} = e_j^t - \sum_{i=1}^{m} B_{i,j} \tilde{r}_i^{t+1}.$$

Once the second task is completed, the message node now applies the soft thresholding operation which only depends on values it has access to.

$$e_j^{t+1} = \eta^S(\tilde{e}_j^{t+1}, \tau\nu).$$

## C   Associative memories based on sparse coding

In this section, we explore another natural model for the dataset (signals) to be stored on an associative memory. Many signals in the real life can be expressed as combinations of a small number of signals from a set of representative signals, namely a *dictionary*. In particular, assuming that $\mathbf{D} \in \mathbb{R}^{n \times m}$ denotes the dictionary with $m$ representative signals from $\mathbb{R}^n$ with $m \geq n$, a valid message (signal) to be stored has the form $\mathbf{y} = \mathbf{Dx}$, where $\mathbf{x} \in \mathbb{R}^m$ is a sparse vector. Given a set of valid messages $\mathbf{y}_1, \mathbf{y}_2, \ldots, \mathbf{y}_N$, finding the dictionary or coding matrix $\mathbf{D}$ and sparse coefficient vectors $\mathbf{x}_1, \mathbf{x}_2, \ldots, \mathbf{x}_N$ such that $\mathbf{y}_i$ is approximated by $\mathbf{Dx}_i$ for every $i \in [N]$, is known as the *dictionary learning* problem. This problem was first studied by Olshausen and Field in [24]. Recently this problem has received a great amount of attention where given certain incoherence assumptions on the dictionary $\mathbf{D}$ and suitable sparsity bounds on the coefficient vectors, provable algorithms for the dictionary learning problem have been proposed in [3, 1, 2] and references therein.

Along this direction, we here comment on how the sparse coding approach can be utilized to construct associative memories. Learning the dictionary $\mathbf{D}$ constitutes the learning phase for the associative memory. Towards this, one can employ one of the dictionary learning algorithms proposed in [3, 1, 2]. These algorithms work under the assumption that the dictionary $\mathbf{D} \in \mathbb{R}^{n \times m}$ has coherence parameter $\mu \leq \frac{c}{\sqrt{n}}$. Moreover, the valid messages are assumed to have coefficient vectors with sparsity at most $O(\sqrt{n})$ with nonzero coefficients taking absolute values that lie in a small enough interval bounded away from zero.

For the recovery phase, we are interested in recovering the coefficient vector (and thus the message vector) from $\mathbf{z} = \mathbf{Dx} + \mathbf{e}$, where $\mathbf{e}$ is a noise vector. Assuming $\mathbf{e}$ has its entries distributed according to a sub-gaussian distribution, one can solve the Lasso problem $\min_{\mathbf{x} \in \mathbb{R}^m} \frac{1}{2n} \|\mathbf{y} - \mathbf{Dx}\|_2^2 + \lambda_n \|\mathbf{x}\|_1$ to recover $\mathbf{x}$ provided that the dictionary matrix $\mathbf{D}$ satisfies some additional assumptions [28, Theorem 1]. In [28], Wainwright bounds the coordinate-wise error between $\mathbf{x}$ and the solution of the Lasso problem mentioned above by a function of $\lambda_n$. Therefore, if nonzero values in a coefficient vector are picked from a discrete set such that any two elements of the discrete set are at least $2g(\lambda_n)$ (a quantity that depends on the regularization parameter $\lambda_n$) far away, then solving Lasso recovers the coefficient vector $\mathbf{x}$ and subsequently the signal $\mathbf{y} = \mathbf{Dx}$ exactly.