[Reviews · NeurIPS 2015]

Submitted by Assigned_Reviewer_1

Mathematically, the paper "studies the following problem. Assuming in a data set the data points are of form y = x + e where e is a sparse noise, and x lies in a lower dimensional subspace. The goal is to design an algorithm that recover x in a neurally plausible way.

The solution of the paper given is the following. Suppose the network can learn a matrix B which is in the null space of x's, that is, B is such that for any x, Bx = 0. Then to recover x, one only needs to first compute By = Be and then recover e using sparse recovery and then recovery e.

I don't see any guarantee that why such B can be founded. Also the learning algorithm only finds incoherent B at most but not RIP. The two main theorems are not proved formally. Results in Section 3 and 5 are almost well known.
Summary: The paper is not well written enough and confusing in the neural feasibility part (for example, it seems that authors think that an algorithm is neurally feasible if it only uses matrix vector inner products..) Section 3 and 5 seem to contain well known results.

Submitted by Assigned_Reviewer_2

The authors consider the problem of associative memory storage and recalling using a sparse recovery model. More storing the memory, they learn a set of linear constraints that the vectors must satisfy, and establish capacity bounds using the null space property and restricted isometry property from compressed sensing. The memory recall assumes that the error signal is sparse, and use sparse recovery methods to reconstruct the error and retrieve the memory.

Quality/Clarity: The paper is unfortunately poorly written. Theorems 1 and 2 (the central claims) are stated without proofs, and they do not connect theorem 3 and Proposition 4 to them, which would clarify how their results hold. It is not clear how exponential capacity is achieved.

Originality: As the authors acknowledge, their idea and algorithm of choice for finding the right null space is not novel [13].

The neural feasibility of the iterative soft thresholding algorithm using Bregman divergence is known in the literature (See e.g. Hu, Genkin and Chklovskii, Neural Computation 2012). The authors devote more than a page in their manuscript for this without acknowledging that.

Maybe I miss something, but I was also not particularly impressed with Theorem 3 on the null space property. I thought that more tight (logarithmic) results can be obtained there similarly to proposition 4 for the RIP, but maybe I miss something.

Significance:

While the problem is interesting the authors do not make comparisons with the recent paper [13] that deals with the same problem. They state that the results in [13] are less robust, but do not show numerical comparisons in their figure. The y-label in Fig 2 seems also to be the opposite from what intended, the probability of failure should decrease with sparsity.
Summary: The authors deal with the problem of associative memory storage and retrieval using ideas from sparse signal recovery. Unfortunately, their main claims are not proven and the novelty compared to other methods is questionable.

Submitted by Assigned_Reviewer_3

- Summary This paper proposes a Hopfield-like model of associative memory based on sparse signal recovery. They learn a set of linear constraints satisfied by all data vectors. i.e., a basis for the null space of the data's span. This basis is analogous to compressive sensing measurements. They then show that, for certain assumptions on the dataset, they end up with a null space basis that satisfies sufficient conditions for sparse recovery. Then they show that there is a neurally-plausible algorithm by which the data vectors can be reconstructed from the null-space basis.

- Quality The experiments illustrate the approach's feasibility. It's nice to see the results for the different choices of dataset and noise model. The rest of the paper is also convincingly argued.

- Clarity The paper is well-explained. However, it has many distracting typos. It would benefit from additional proofreading.

- Significance Promisingly, the recall algorithm proposed here is neurally plausible. It might motivate further research in computational neuroscience or neural net modeling along these lines.

- Minor 1. There is a typo in the RIP equation (2), on the RHS it should be (1 + delta). 2. The word "the" is missing in the second to last sentence of the first paragraph of 3.1. (line 190) 3. The word "probability" is duplicated in the last sentence of the last paragraph of 3.1. (line 247) 4. The last line on page 5 is missing the word "the" (line 269). 5. Line 280, this sentence is not grammatical.
Summary: This paper proposes an interesting associative memory model based on sparse signal recovery. The recall algorithm is neurally-plausible and may motivate further research in this area.

Submitted by Assigned_Reviewer_4

This paper builds on the idea of structured pattern sets in associative memories allowing a much larger capacity.

This idea was first developed by Venkatesh and by Biswas in the 1980s: should perhaps be mentioned. In intro, might also want to mention how randomization in recall improves performance of Karbasi et al. [13], in follow-on work by Karbasi et al.

(Karbasi et al., Neural Computation, 2014). This also raises the question whether the present work would benefit from randomness.

The basic idea here is to have pattern sets that essentially yield a sparse recovery problem,

that can then be solved using message-passing algorithms.

For example the RIP.

Both the learning and recall algorithms look to be correct, as stated.

Likewise characterization of size of pattern sets, i.e. capacity.

Can anything be said about biological plausibility in terms of anatomical networks?
Summary: The work is a solid way to consider structured pattern sets in associative memories, and by making connections with sparse recovery (and their iterative algorithms), new insights are achieved.

Work appears correct.

Submitted by Assigned_Reviewer_5

Associative memory via a Sparse Recovery Model -------------------------------------------------- This paper builds an associative memory using message passing in a random graph setting.

For an appropriately chosen set of constraints one can create an associative memory that seems to perform well using only local computations.

I think the task is valid and the solution is interesting.

I would have preferred more experimental work, so I am recommending a weak accept.

More technical comments: ----------------------- Minor quibble - on line 46 - please refer to 1.22^n as exp(0.2*n).

It's just more consistent with the rest of your notation in the rest of the paper

On line 95, you mention that your model is robust against adversarial error patterns but then don't mention it again.

Can you please go into more detail about this claim or drop it entirely?

Regarding equation 20, you casually mention that it is a neurally feasible algorithm since it involves only matrix vector multiplications.

However matrix B appears twice in the equation, which seems to imply that there is a non-local dependency in your algorithm. (i.e. if one were to change a tweak a single parameter in the B matrix then one would have to modify your neural implementation in two, non local places).

This makes it hard to see how such an algorithm could be learnt.

If you do have any ideas n how this could be learnt then it would be great to add them, as it would really strengthen the paper.

For your experimental section I would like to see more evidence showing that scaling behaviour that you predict does indeed show up in practice as you scale the network out.

(i.e. iterative soft thresholding doesn't change the theoretical analysis.)

It would also be great to see information theoretic performance measures.

e.g. how does the mutual information between the initial cue and true pattern compare with the mutual information between the recalled pattern and the true pattern?

Please consider also citing: Sterne (2012) - Efficient and Robust Associative Memory from a Generalized Bloom Filter which also does associative memory using a random graph with message passing.

If you think the paper is very different then I won't push the point, but it seemed relevant to me.
Summary: This paper builds an associative memory using message passing in a random graph setting.

For an appropriately chosen set of constraints one can create an associative memory that seems to perform well using only local computations.

I think the task is valid and the solution is interesting.

I would have preferred more experimental work, so I am recommending a weak accept.

Author Feedback
Author rebuttal: We are grateful to the reviewers for their thorough reading of our submission and the comments they provided for the betterment of this paper. Please find our response below in the order the points were raised by the reviewers.

Assigned_Reviewer_1:

1. Proofs of Thm. 1 and 2. The proof of Thm. 1 follows straight-forwardly from Thm. 3 and Prop. 1. This has been discussed in the techniques section (see the points of Sec.1.1) and beginning of Sec.3. Indeed, the sub-Gaussian dataset has dimension O(n^{3/4}) and every vector in this space satisfy Eq. (6) (see, Thm. 3). Hence, the null-space of this dataset satisfies the premise of Prop. 1. On the other hand, the size of the dataset, restricted to any discrete alphabet is exp(O(n^{3/4}). Similarly Thm. 2 follows from Prop. 2 and 4. We understand that we could be more explicit in this connection (which we will do if the paper is accepted). But, the technical proofs are present in the paper (and are correct).

2. Claims of exponential capacity. Under the assumptions on the model of the dataset, and by the parameters of Thm 3 and Prop. 4, it follows that the dataset stored can be as large as claimed.

3. Novelty compared to reference [13]. We have tried our best to highlight the differences between [13] and our work in the Introduction, especially in Sec.1.1. The only similarity between [13] and our work is that we consider those datasets which belong to a subspace and the learning phase amounts to learning a basis of the orthogonal subspace. Our recall phase relies on sparse recovery algorithms based on RIP, null sparse property or incoherence. On the other hand, the recall phase in [13] relies on peeling algorithm by Luby et al. in coding theory literature. (Also, as noted in the Introduction, it is not clear if the learning phase in [13] gives the basis vectors which meets the requirements to ensure the success of the recall phase used in [13].) Moreover, as specified in our paper, [13] only deal with random errors in messages during the recall phase.

4. Bregman Iterative Algorithm (BIA). Each iteration of the BIA is performed using the iterative soft thresholding algorithm. While the reference reviewer 1 provided (Hu et al.) is relevant, it does not subsume what we do. In particular, the neural structures employed by Hu et al. is different from the one we explain in Appendix~B. We do not claim to be the first one to observe the neural feasibility of BIA. We have presented discussion on neural feasibility for the completeness as it justifies the use of BIA as apposed to other popular algorithms for sparse recovery (e.g., OMP). If the paper is accepted, we will cite Hu et al. (with proper context) when discussing the neural plausibility of BIA.

5. Tightness of Thm. 3. It is possible to choose slightly different values for k and r, and what was provided is an example set of values for these parameters.

6. Correctness of Fig. 2. The y-label in Fig. 2 is correct. We use sparsity of a vector to denote the number of nonzero entries in the vector. Therefore, higher 'sparsity' implies more nonzero entries in a vector. As the number of nonzero entries increase in a vector, it becomes harder to recover the vector from a given number of measurements.

Assigned_Reviewer_4:

7. Adversarial Error Model. Throughout this paper, as in particular explained in Sec. 5, we use an adversarial error model. There is no need to assume a random model for our analysis, as the sparse-recovery literature provides ways to recover arbitrary sparse vectors, as opposed to random ones. This is one advantage compared to [13].

8. Neural feasibility (of Eq. (20)). We clearly state in line 78 what we mean by neural feasibility: "A neurally feasible algorithm ... employs only local computations at the vertices of the corresponding bipartite graph based on the information obtained from their neighboring nodes." Note that Eq. (20) is similar to Eq.(17). The Eq. (17) also involves the matrix B twice. We explain in Appendix~B how one can implement Eq.(17) in a neurally plausible manner. Using similar approach Eq.(20) can be implemented in a neurally plausible manner (without the soft thresholding at message nodes).

Assigned_Reviewer_7:

9. Guarantee of finding incoherent or RIP null-space. Of course there is no guarantee that for an arbitrary dataset a null-space of appropriate properties can be found - and this is the reason we assumed randomized models for the dataset. This is one of the main points addressed in the paper. In the Introduction it is clearly mentioned: "Indeed, for a large class of random signal models, we show that, such constraints must exists and can be found in polynomial time."This point further emphasized in Sec. 1.1, point 1. The reviewer says that "Sec. 3 is almost well-known," but this is exactly what was established in there.

We thank the other reviewers for their very positive reviews. We are going to fix the remaining typos.